# 3,6′- and 1,6′-Dithiopomalidomide Mitigate Ischemic Stroke in Rats and Blunt Inflammation

**DOI:** 10.3390/pharmaceutics14050950

**Published:** 2022-04-27

**Authors:** Yan-Rou Tsai, Dong Seok Kim, Shih-Chang Hsueh, Kai-Yun Chen, John Chung-Che Wu, Jia-Yi Wang, Yi-Syue Tsou, Inho Hwang, Yukyung Kim, Dayeon Gil, Eui Jung Jo, Baek-Soo Han, David Tweedie, Daniela Lecca, Michael T. Scerba, Warren R. Selman, Barry J. Hoffer, Nigel H. Greig, Yung-Hsiao Chiang

**Affiliations:** 1Neuroscience Research Center, Taipei Medical University, Taipei 110, Taiwan; d620101002@tmu.edu.tw (Y.-R.T.); kychen08@tmu.edu.tw (K.-Y.C.); dr.jcwu@gmail.com (J.C.-C.W.); jywang2010@tmu.edu.tw (J.-Y.W.); 2Department of Neurosurgery, Taipei Medical University Hospital, Taipei Medical University, Taipei 110, Taiwan; b101091102@tmu.edu.tw; 3Aevisbio Inc., Gaithersburg, MD 20878, USA; dskim@aevisbio.com; 4Aevis Bio Inc., Daejeon 34141, Korea; ihwang@aevsibio.com (I.H.); yuki@aevisbio.com (Y.K.); gilje620@gmail.com (D.G.); ejo@aevisbio.com (E.J.J.); 5Drug Design & Development Section, Translational Gerontology Branch, Intramural Research Program National Institute on Aging, NIH, Baltimore, MD 21224, USA; shih.hsueh@nih.gov (S.-C.H.); tweedieda@grc.nia.nih.gov (D.T.); daniela.lecca@nih.gov (D.L.); mike.scerba@nih.gov (M.T.S.); 6The Ph.D. Program for Neural Regenerative Medicine, College of Medical Science and Technology, Taipei Medical University, Taipei 110, Taiwan; 7Department of Surgery, College of Medicine, Taipei Medical University, Taipei 110, Taiwan; 8Taipei Neuroscience Institute, Taipei Medical University, Taipei 110, Taiwan; 9Research Center for Biodefence, Korea Research Institute of Bioscience and Biotechnology, Daejeon 305-806, Korea; bshan@kribb.re.kr; 10Department of Neurological Surgery, Case Western Reserve University, Cleveland, OH 44106, USA; warren.selman@uhhospitals.org (W.R.S.); bjh82@case.edu (B.J.H.); 11University Hospitals Cleveland Medical Center, Cleveland, OH 44106, USA

**Keywords:** ischemic stroke, 3,6′-dithiopomalidomide, 1,6′-dithiopomalidomide, pomalidomide, cereblon, inflammation, TNF-α, immunomodulatory imide drugs (IMiDs)

## Abstract

(**1**) **Background:** An important concomitant of stroke is neuroinflammation. Pomalidomide, a clinically available immunomodulatory imide drug (IMiD) used in cancer therapy, lowers TNF-α generation and thus has potent anti-inflammatory actions. Well-tolerated analogs may provide a stroke treatment and allow evaluation of the role of neuroinflammation in the ischemic brain. (**2**) **Methods:** Two novel pomalidomide derivatives, 3,6′-dithiopomalidomide (3,6′-DP) and 1,6′-dithiopomalidomide (1,6′-DP), were evaluated alongside pomalidomide in a rat middle cerebral artery occlusion (MCAo) stroke model, and their anti-inflammatory actions were characterized. (**3**) **Results:** Post-MCAo administration of all drugs lowered pro-inflammatory TNF-α and IL1-β levels, and reduced stroke-induced postural asymmetry and infarct size. Whereas 3,6′- and 1,6′-DP, like pomalidomide, potently bound to cereblon in cellular studies, 3,6′-DP did not lower Ikaros, Aiolos or SALL4 levels—critical intermediates mediating the anticancer/teratogenic actions of pomalidomide and IMiDs. 3,6′-DP and 1,6′-DP lacked activity in mammalian chromosome aberration, AMES and hERG channel assays –critical FDA regulatory tests. Finally, 3,6′- and 1,6′-DP mitigated inflammation across rat primary dopaminergic neuron and microglia mixed cultures challenged with α-synuclein and mouse LPS-challenged RAW 264.7 cells. (**4**) **Conclusion:** Neuroinflammation mediated via TNF-α plays a key role in stroke outcome, and 3,6′-DP and 1,6′-DP may prove valuable as stroke therapies and thus warrant further preclinical development.

## 1. Introduction

In 2019, according to the datasheet from the World Health Organization, stroke was the second major leading cause of death, responsible for approximately 11% of total deaths and the leading cause of long-term neurological disability in the world [1]. Two specific types of stroke mechanisms cause the vast majority of strokes: ischemic and hemorrhage stroke. Ischemic stroke accounts for 87% of all strokes in the US; hemorrhagic stroke represents approximately 10 to 15%, although its frequency is higher in Asia [2,3]. As ischemic strokes are more prevalent, they have been the main focus of most drug trials [3,4,5]. People who survive after stroke often experience long-term neurological disabilities. A particularly serious consequence of cerebral ischemic stroke is dementia, which is also associated with significant long-term disability [2].

The common cause of brain ischemic injury is pathophysiological thrombosis and thromboembolism giving rise to cerebral arterial occlusion. The two most common sources of emboli are the left side cardiac chambers and large arteries [6]. The series of neurochemical processes that are initiated by transient or permanent focal cerebral ischemia are referred to as the ischemic cascade; the ischemic cascade is a highly heterogeneous phenomenon. However, it can be summarized as cellular bioenergetic failure due to focal cerebral hypoperfusion, excitotoxicity, oxidative stress, blood–brain barrier (BBB) disruption, microvascular injury, loss of ion homeostasis, post-ischemic inflammation and death of neurons, glia and endothelial cells.

### 1.1. Post-Ischemic Inflammation

Inflammation leads to stroke-related brain injury. However, individual components of the inflammatory cascade can be beneficial depending on the stage of tissue injury, the extent of the response and whether the inflammatory component activates neuroprotective pathways [2,7]. The inflammatory response involves many different cell types, inflammatory mediators and extracellular receptors. First, microglia and astrocytes are activated by reactive oxygen and nitrogen species (ROS, RNS). Such glial cells are capable of secreting inflammatory factors such as cytokines/chemokines and inducible nitric oxide synthase (iNOS) [4,5,8]. In rodent cerebral ischemia models, infiltrating neutrophils generate iNOS that produces toxic amounts of NO [4,5,6,7]. Additionally, ischemic neurons express cyclooxygenase 2 (COX2), an enzyme that mediates ischemic brain injury by producing superoxide and toxic prostanoids [9]. Microglia are the macrophages of the brain and play a crucial role in the central nervous system [8]. After ischemia, microglia transform from a quiescent to an activated state, develop a phagocytic phenotype and release a variety of substances that are cytotoxic and/or cytoprotective. Within 4–6 h post-ischemia, circulating leukocytes attach to vessel walls and migrate into the brain with subsequent release of more pro-inflammatory mediators and secondary injury, impacting potential recovery-based tissue in the penumbra [8].

Key cytokines related to inflammation in stroke are tumor necrosis factor-α (TNF-α), interleukin (IL)-1, IL-6 and IL-10 [10,11,12]. Whereas IL-1 is pro-inflammatory, TNF-α has multiple functions and may impact apoptosis or survival through different pathways [4,5,6,7]. Elevated production of pro-inflammatory cytokines and lower levels of the anti-inflammatory IL-10 are related to larger infarctions and poorer clinical outcomes [13]. Chemokines are important for cellular communication and inflammatory cell recruitment. In addition to chemotactic properties, chemokines were found to directly impact BBB permeability [7]. Inflammatory cells also generate ROS and RNS and produce matrix metalloproteinases, inducing more damage to the ischemic brain [8].

### 1.2. Pomalidomide and Its Mechanism

Thalidomide was found to inhibit angiogenesis in 1994 but was initially marketed as a sedative in the late 1950s. Thalidomide was widely used as a treatment for morning sickness in pregnant women in Europe [14]. However, publications by McBride in Australia and Lenz in Germany documented limb and bowel malformations in children born to mothers exposed to thalidomide during pregnancy in 1962 [15]. Fortunately, thalidomide was not FDA-approved as a sedative in the US due to neurotoxicity concerns, and its widespread international use was halted during the early 1960s.

Thalidomide and its second and third generation analogs, lenalidomide and pomalidomide, were later developed as immunomodulatory imide drugs (IMiDs) that display a multitude of biologic effects on cytokine and cell-mediated responses, and are effectively used in the treatment of multiple myeloma [16]. Pro-angiogenic cytokines and bone marrow vascularization, key elements in multiple myeloma progression, were prospective targets that could be exploited through the anti-angiogenic properties of thalidomide [16], and more effectively still by pomalidomide with more potent pro-inflammatory cytokine lowering action [16,17]. All three thalidomide analogs [18,19] are structurally similar and share an α-(isoindolinone-2-yl)-glutarimide core structure [20]. Key elements within this structure, particularly the unsubstituted glutarimide moiety, support binding of these agents to a surface domain on the protein cereblon [21,22]. Cereblon interacts with other proteins to form a DNA damage-binding-protein cereblon E3 ligase complex that, when bound to thalidomide or similar analogs, results in the ubiquitination and proteasomal degradation of key proteins [23]. The best-known ones that are substrates for cereblon–IMiD binding are the zinc finger family proteins Ikaros, Aiolos and SALL4, and such substrate binding results in the known anticancer and teratogenic actions of thalidomide and similar drugs, but not their anti-inflammatory effects [16,21,24]. 

Focusing on their cytokine lowering actions, IMiDs have more recently been evaluated as a new treatment for neurological disorders with an inflammatory and neuroinflammatory component [25,26,27,28,29]. In the present study, we evaluated the actions of two novel pomalidomide derivatives, 3,6′-dithiopomalidomide and 1,6′-dithiopomalidomide (1,6′- and 3,6′-DP), in the middle cerebral artery occlusion (MCAo) model of stroke, and characterized their actions to assess whether these agents might represent candidates worthy of development as new therapeutics. 

## 2. Materials and Methods

### 2.1. Drug Synthesis

Pomalidomide (QB-4271, Combi-Blocks, San Diego, CA, USA) was recrystallized prior to use to ensure high purity. 3,6′- and 1,6′-DP were synthesized from pomalidomide according to Scerba et al. [30]. Details and chemical characterization are provided in the Appendix A. 3,6′-DP and 1,6′-DP were freshly prepared immediately prior to use across all studies. 

### 2.2. Stroke Studies

#### 2.2.1. MCAo Procedure

Transient MCAo was performed in male Sprague Dawley rats (175–200 g weight, BioLASCO Taipei, Taiwan) in accordance with the National Institutes of Health (DHEW publication 85–23, revised, 1996) under the approval of the Animal Care and Use Committee of the National Defense Medical Center, Taipei, Taiwan (protocol: IACUC-20-043). Animal numbers and experimental measures were selected based upon our prior studies [25], and the transient MCAo procedure and drug treatment are described in the Appendix A [31,32]. All MCAo-challenged animals demonstrated prominent motor bias contralateral to the lesion side, and animal numbers utilized were based on the variance of data and differences between treatment and control groups evident in both our prior studies and those of others [32,33,34]. 

In this initial evaluation of 3,6′-DP and 1,6′-DP in a rodent model of cerebral ischemia, studies were performed in male rats only to appraise a signal of efficacy. Gender differences are known in the response of rodents to MCAo, as well as to other brain insults [35,36]. In this regard, it is well established that stroke incidence in young to middle-aged adults is lower in women than men. This trend reverses later in life, particularly when women enter their postmenopausal stage [37]. Likewise, young female rodents appear to be more resistant to ischemic brain injury than age-matched males [38,39]. Studies in ovariectomized rodents have demonstrated that estrogens provide significant neuroprotection to counter ischemic stroke damage in female animals [38,39,40]. In contrast, estrogen is reported to enlarge infarct volume in aged female rodents. In summary, these studies suggest that estrogen protects the younger female brain but exacerbates the damage in the aging brain following ischemic stroke [41,42]. To avoid the confounds associated with estrogen generation in young female rats, ovariectomizing animals or aging them to a postmenopausal state, we performed our initial studies, reported herein, in young adult rats. 

Rats were divided into five groups: sham (without MCAo) and, for those with MCAo, randomly (computer generated) between vehicle MCAo, 3,6′-DP, 1,6′-DP and pomalidomide MCAo groups. Following 60 min MCAo and 30 min reperfusion, groups received i.p. injections of vehicle (DMSO), 3,6′-DP, 1,6′-DP or pomalidomide, respectively (equimolar to pomalidomide 20 mg/kg). 

#### 2.2.2. Injured Cerebral Lesion Area Assessment and Neurological Deficit Scores

Infarct size was measured 24 h after MCAo based on Shen et al. [32] by an observer blinded to treatment groups, as described in Appendix A [32,34]. Motor/neurological deficits were assessed using the body asymmetry test of Borlongan and Sanberg [33,34] (Appendix A). The maximum impairment in body swing in MCAo rats is 20 contralateral turns/20 trials, with animals demonstrating asymmetric swing ipsilateral to the MCAo [33,34], and uninjured animal showing a value of 10 (equal numbers of left/right turns).

#### 2.2.3. Cytokine Evaluation in MCAo-Challenged Rats

Plasma was isolated for quantification of post-treatment cytokine/chemokine levels across groups (Milliplex MAP Rat Cytokine/Chemokine Kit; Millipore, Billerica, MA, USA) (Appendix A).

### 2.3. Cereblon Binding/Neo-Substrate Studies in Human Cell Cultures

Quantitative evaluation of cereblon binding was performed utilizing bead-based AlphaScreen technology (BPS Bioscience, San Diego, CA, USA, #79770). Pomalidomide, 3,6′-DP or 1,6′-DP (0.01 and 100 μM) was incubated with reaction mixtures containing cereblon/DDB1-CUL4A-Rbx1 (Cullin 4a-Ring-box protein 1) complex (12.5 ng) and bromodomain-containing protein 3 (BRD3) (6.25 ng) in an Optiplate 384-well microplate (PerkinElmer, Waltham, MA, USA, #6007290). After 0.5 h, AlphaLISA anti-FLAG Acceptor and Alpha Glutathione Donor beads (PerkinElmer) were added and incubated (1 h, RT, for each agent). Alpha-counts were, thereafter, recorded on a Synergy Neo2 microplate reader (BioTek, Shoreline, WA, USA). Relative activity was determined as the highest value (established as 100%) and the lowest value (0%) following subtraction of “blank values” from all readings.

Pomalidomide, 3,6′-DP and 1,6′-DP (0.01–1 μM for 24 h) actions on the neo-substrates Aiolos and Ikaros were characterized in human MM1.S (multiple myeloma) cells (ATCC, Manassas, VA, USA). Drug actions on the neo-substrate SALL4 were evaluated in H9 hESC (human embryonic stem) cells (#WA09: WiCell Research Institute, Madison, WI, USA). Neo-substrate protein levels were quantified by Western blot (Appendix A) [28]. The primary antibodies probed included: anti-Aiolos (cat#15103; 1:1000 dilution; Cell Signaling Technology, Danvers, MA, USA), anti-Ikaros antibody (cat#9034; 1:1000 dilution; Cell Signaling Technology), anti-SALL4 (SC101147; 1:1000; Santa Cruz Biotechnology, Dallas, TX, USA), anti-β-actin (CST #3700; 1:5000; Cell Signaling Technology) and GAPDH was used as an internal control (cat#ab8245; 1:5000 dilution; Abcam, Waltham, MA, USA). HRP conjugated secondary antibodies were Goat anti-rabbit IgG (ThermoFisher Scientific, Waltham, MA, USA) and (ii) Goat anti-mouse IgG. Antigen–antibody complexes were detected using enhanced chemiluminescence (iBright CL1500, ThermoFisher Scientific) (Appendix A).

### 2.4. Anti-Inflammatory Animal Studies

#### Lipopolysaccharide (LPS) Inflammatory Challenge

Male Fischer 344 rats (150 g, Charles River Laboratories, Wilmington, MA, USA) were randomly assigned across vehicle, 1,6′-DP and 3,6′-DP (29.5 mg/kg, i.p.) groups and challenged and challenged 60 min later with LPS (1 mg/kg, Sigma, St Louis, MO, USA *E.coli* O55:B5 in saline (0.9%), 0.1 mL/kg i.p.) or vehicle (saline) in accord with National Institute on Aging, NIH, Baltimore, MD, USA, approved Animal Care and Use Committee protocol No. 331-TGB-2024. Animals were euthanized 4 h later and plasma and brain (hippocampus) samples were harvested and stored (−80 °C). Protein concentrations were determined by Bicinchoninic acid assay (ThermoFisher Scientific). TNF-α, IL-6, IL-1β and IL-10 protein levels were quantified by ELISA (Mesoscale Discovery, Meso Scale Diagnostics, Rockville, MD, USA).

### 2.5. Cellular Inflammatory and Survival Studies

#### 2.5.1. 1,6′-DP and 3,6′-DP Actions in Primary Dopaminergic Neurons + Microglia Challenged with α-Synuclein

Rat dopaminergic neurons and microglia were isolated and maintained in culture (Appendix A). On day 7 in culture, cells were preincubated (1 h) with 1,6′-DP, 3,6′-DP or vehicle (0–30 μM), and then exposed to oligomeric α-synuclein (250 nM, 72 h, n = 6/group: human recombinant α-synuclein 1-140aa rPetide, Watkinsville, GA, USA).

Quantification of the following was undertaken: (i) dopaminergic neuron survival (number of tyrosine hydroxylase (TH)-positive neurons across conditions versus control (vehicle/without α-synuclein) condition. (ii) Total dopaminergic neuron neurite network (length of TH-positive neurites). (iii) Total microglia activation (area of microglial cells: μm^2^ of OX-41 staining). (iv) TNF-α protein level (rat TNF-α ELISA, Abcam, ab46070). 

#### 2.5.2. Drug-Induced Anti-Inflammatory Actions in LPS Activated RAW 264.7 Cells

RAW 264.7 cells were preincubated (1 h) with vehicle or drugs (0.6–60 μM, n = 4/group) and then challenged with LPS (60 ng/ mL for 24 h, Sigma, St Louis, MO, USA, serotype 055:B5). Cell viability was measured by CellTiter-Blue Cell Viability assay (cat#G8081, Promega, Madison, WI, USA). Nitrite ions (NO_2_^−^) were quantified by Nitrate/Nitrite Fluorometric Assay Kit (Abnova, Taipei, Taiwan, cat#KA1344) as a surrogate of NO levels, as nitrite is a stable non-volatile breakdown product of NO. TNF-α levels were measured by Biolegend ELISA MAX Set Delux ELISA (Biolegend, San Diego, CA, USA, #430904) (Appendix A).

### 2.6. FDA IND-Enabling Studies

3,6′-DP and 1,6′-DP were evaluated for genotoxicity liability by the classical bacterial reverse mutation (Ames) assay to appraise mutagenicity (5–100 µM), and by the in vitro chromosomal aberration assay to assess chromosomal damage (125–500 µg/mL). Furthermore, they were evaluated for cardiac safety by hERG patch clamp assay (1–30 µM) (Appendix A).

### 2.7. Statistical Analyses

Number of animals/samples evaluated across studies was determined by incorporating the outcome measures from our prior studies [26,28,29] and a power analysis [43]. Data were assessed for presence of statistical outliers (ROUT) and any observed were removed. Data were assessed for normality (Shapiro–Wilk test). If normally distributed, they were assessed using an Ordinary ANOVA, followed by a Dunnett’s multiple comparison (Dunnett’s *t* test), or a Brown–Forsythe and Welch’s ANOVA test followed by the Dunnett’s T3 multiple comparison test. For not normally distributed data, a non-parametric Kruskal–Wallis test followed by a Dunn’s multiple comparison test were utilized. All graphs were plotted using Prism. 

## 3. Results

### 3.1. Stroke Studies In Vivo

Quantitation of brain infraction volume showed marked reduction in all groups (Figure 1B,C); the MCAo + 3,6′-DP group was 7.41 ± 1.06%, the MCAo + 1,6′-DP group was 6.37 ± 1.43% and the MCAo + pomalidomide group was 5.62 ± 2.25%, which were all lower than the MCAo + vehicle group (12.49 ± 1.91%) (Figure 1C). Body asymmetry was also markedly reduced by all three drug treatments versus MCAo vehicle animals. The MCAo + 3,6′-DP group was 15.5 ± 0.64%, the MCAo + 1,6′-DP group was 12.5 ± 1.84% and the MCAo + pomalidomide group was 13.75 ± 0.85%, which all were lower than the MCAo + vehicle group (19.25 ± 0.25%) (Figure 1D). 

Ischemic stroke triggers a vigorous inflammatory response initiated within minutes and persisting for days [45,46]. After ischemic stroke, a secondary immune response occurs that includes glial activation and release of cytokines and chemokines. To study the effects of pomalidomide and analogs in ischemic-stroke-induced inflammation, we investigated cytokine levels after post-treatment of pomalidomide and analogs in MCAo.

Minutes to hours after stroke, the acute phase of stroke leads to production of pro-inflammatory cytokines, particularly IL-1β and TNF-α, which propagate the neuroinflammatory response through activation of danger signals [7,45]. To investigate the regulation of pomalidomide and analogs on global inflammatory responses after ischemic stroke, we quantified pro- and anti-inflammatory cytokine levels. After brain ischemia, levels of the pro-inflammatory IL-1β and TNF-α were elevated in the MCAo + vehicle group (155% and 187% of sham control levels, respectively), whereas anti-inflammatory IL-10 levels were not significantly changed (88% of sham values). Notably, pomalidomide and analogs significantly reduced the ischemia-induced pro-inflammatory cytokines, with all analogs lowering IL-1β and TNF-α to below sham control levels. Anti-inflammatory IL-10 levels were unchanged by pomalidomide and 1,6′-DP in MCAo-challenged animals versus the MCAo + vehicle group, but were notably significantly elevated in MCAo + 3,6′-DP animals (212%, Table 1).

### 3.2. Cereblon Studies

3,6′-DP, 1,6′-DP and pomalidomide potently bind cereblon, but 3,6′-DP does not lower downstream neo-substrates Aiolos, Ikaros and SALL4. Previous research has demonstrated that cereblon is a direct protein target for the immunomodulatory, antiproliferative and teratogenic activities of thalidomide and analogs. Cereblon binding induces ubiquitination of the lymphoid transcription factors Ikaros and Aiolos and also SALL4 via the cereblon E3 ubiquitin ligase for proteasomal degradation [16,24,27]. We therefore evaluated the concentration-dependent binding of pomalidomide, 3,6′-DP and 1,6′-DP to cereblon, and found that all three potently bound (IC_50_ values 2.37, 0.94, 1.52 μM, respectively; Figure 2A,B). As expected, pomalidomide triggered degradation of Aiolos (Figure 2C), Ikaros (Figure 2D) and SALL4 (Figure 2E), significantly lowering their expression concentration-dependently in human MM1S (0.01–1 μM) or H9 cells (3 μM). In contrast, a higher concentration of 1,6′-DP was required to induce a significant decline in either Aiolos or Ikaros. Importantly, no significant decline in the protein expression of Aiolos, Ikaros or SALL4 was evident with 3,6′-DP. This indicates that despite similar cereblon binding, 3,6′-DP, and to a lesser extent 1,6′-DP, did not trigger ubiquitination and reduction in the key downstream neo-substrates Aiolos, Ikaros and SALL4.

### 3.3. LPS-Mediated Inflammation In Vivo

Consequent to 3,6′-DP- and 1,6′-DP-mediated lowering of elevated plasma pro-inflammatory cytokine levels induced by MCAo (Table 1), we evaluated the ability of these drugs to mitigate systemic and brain inflammation in a classical systemic LPS model. Illustrated in Figure 3, systemic LPS administration induced marked elevations of plasma pro-inflammatory cytokines TNF-α, IL-6 and IL-1β, together with anti-inflammatory IL-10. In brain hippocampal tissue similarly evaluated at 4 h post challenge, TNF-α and IL-6 were significantly elevated.

3,6′-DP and 1,6′-DP attenuated the levels of LPS-induced pro-inflammatory cytokines, with 3,6′-DP achieving significant reductions in TNF-α, IL-6 and IL-1β in plasma as well as of TNF-α in the brain. 1,6′-DP, likewise, significantly lowered LPS-induced levels of TNF-α and IL-6 in plasma and of TNF-α in the brain, with reductions being, in general, less that those achieved for 3,6′-DP. For both agents, declines in pro-inflammatory cytokines were achieved without impacting anti-inflammatory IL-10 levels (Figure 3D).

### 3.4. Cellular Studies on Inflammation and Survival

Consequent to the anti-inflammatory actions of 3,6′- and 1,6′-DP across two rodent models, and significant reductions in infarct volume in treated MCAo-challenged animals, we evaluated the effects of both candidate drugs in cellular models involving inflammation and cellular dysfunction/loss. Specifically, we evaluated these effects in primary dopaminergic neurons + microglia challenged with α-synuclein (Figure 4) and RAW 264.7 cells challenged with LPS (Figure 5).

In the first study, primary co-cultures of dopaminergic cells were preincubated with either 1,6′- or 3,6′-DP (3–30 µM for 1 h) before a 72 h challenge with oligomeric α-synuclein. α-Synuclein-challenged cells in the presence of the vehicle (DMSO) displayed significant reductions in the survival of neurons and their neurite networks (Figure 4A,B, respectively), and significant elevations in their levels of activated microglial cells and TNF-α protein (Figure 4C,D, respectively). The actions of the 3,6′- and 1,6′-DP were dose-dependently evaluated to mitigate α-synuclein actions across these neuronal and inflammatory measures.

Whereas 3,6′-DP and 1,6′-DP attenuated the α-synuclein-induced loss of dopaminergic cells (Figure 4A), only 3,6′-DP reduced the neurite network loss (Figure 4B). Additionally, 3,6′-DP but not 1,6′-DP reduced levels of α-synuclein-induced activated microglia (Figure 4C) and both agents lowered secreted TNF-α protein levels (Figure 4D).

In mouse RAW 264.7 cells challenged with LPS (60 ng/mL) to induce a submaximal elevation in both nitrite (NO^2−^) and TNF-α levels, 3,6′-DP concentration-dependently (1–60 µM) reduced levels of NO^2−^ and TNF-α. The lowering effects of 1,6′-DP on NO^2−^ attained statistical significance at 10 µM with no further reductions observed at 30 and 60 µM. Treatment with 1,6′-DP (30–60 µM) induced significant reductions in the levels of TNF-α protein (Figure 5). 1,6′-DP and 3,6′-DP were associated with a moderate but statistically significant level of cellular toxicity at concentrations of 30–60 µM (not shown).

### 3.5. Essential FDA Requirements for Human Pharmaceuticals

To determine whether 3,6′-DP and 1,6′-DP meet fundamental FDA requirements for future consideration as potential human drug candidates, they were evaluated for genotoxicity liability by the classical bacterial reverse mutation (Ames) assay to appraise mutagenicity, and by the in vitro chromosomal aberration assay to assess chromosomal damage. Additionally, they were evaluated for cardiac safety by the hERG patch clamp assay. All these selected assays are in line with FDA guidance information relating to requirements for human pharmaceuticals. 

As detailed in the Appendix A, chromosomal damage was assessed in human peripheral blood lymphocytes at different times following incubation of compounds in the presence/absence of rat liver S9 extract (to support drug metabolism and evaluation of potential genotoxic metabolites). Thereafter, quantification of chromosome alterations was performed including gaps, breaks (i.e., deletions/displacements) and exchanges, and cyclophosphamide and ethyl methanesulfonate were used as positive controls. Neither 3,6′-DP nor 1,6′-DP (125 μg/mL) produced significant chromosomal aberrations across measures (Appendix A).

Quantitatively evaluated in a standard AMES Fluctuation test for mutagenicity against five different well-characterized, genetically modified bacterial strains in the presence/absence of rat liver S9 extract, 3,6′- and 1,6′-DP induced levels of bacterial growth similar to those of the drug vehicle controls (Appendix A), indicative of a lack of mutagenicity. All positive controls induced a high level of mutagenicity across the bacterial assays.

The effects of 1,6′- and 3,6′-DP (1–30 µM) were assessed on hERG-HEK263 cells by patch clamp analysis of IKr channel conductance. 1,6′- and 3,6′-DP induced maximal inhibitions of 5% and 27% at 30 µM (Appendix A), indicative of hERG channel safety. Propafenone (10 µM) was used as a positive control and induced 85% channel inhibition.

## 4. Discussions

After cerebral ischemia, there is consequent evoked BBB damage, oxidative stress, inflammatory responses, neuronal death and neurological dysfunction. An important concomitant effect of stroke in both humans and preclinical models is neuroinflammation, especially in the penumbra region [3,4,5,6,7,8,10,11,12]. The clinically approved IMiDs, thalidomide and pomalidomide, exhibit potent anti-inflammatory actions [27]. We thus studied the effects of two novel pomalidomide derivatives, 3,6′- and 1,6′-DP, in a preclinical model of stroke, the MCAo model. 

One of the most significant symptoms of stroke is sensorimotor asymmetry, which can be assessed by several behavioral tests, but a particularly sensitive assessment tool is the elevated body swing test of Borlongan et al. [33,34], where a highly significant correlation between volume of infarction induced by proximal MCAo and body asymmetry has been reported. A wealth of studies by Borlongan and colleagues have evaluated drug classes to mitigate this (including agents to impact neuroinflammation). The value of this useful tool has been confirmed by numerous studies, as reviewed by Shen and Wang [48]. Both 3,6′-DP and 1,6′-DP reduced postural asymmetry and infarct size when given after MCAo at a dose of translational relevance, as compared to thalidomide and clinical analogs. The pro-inflammatory cytokines TNF-α and IL1-β were reduced in plasma from stroke animals, whereas anti-inflammatory IL-10 was elevated for 3,6′-DP. Although both 3,6′-DP and 1,6′-DP bound to human cereblon, 3,6′-DP did not activate processes resulting in the degradation of Ikaros, Aiolos and SAL4 that are central in anticancer and teratogenicity actions of IMiDs [16,17]. Neither 3,6′-DP nor 1,6′-DP had any unfavorable actions in the mammalian chromosome aberration assay and the AMES fluctuation assay, key assessments of potential genotoxicity. Additionally, effects of the two compounds on hERG channels were minimal. Finally, we cross-validated the anti-inflammatory actions of both agents in a classical LPS-induced pro-inflammatory cytokine rat assay that demonstrated the ability of systemic drug administration to mitigate both systemic and brain markers of inflammation, and we conducted assays using primary neuronal and glial cells and RAW 264.7 cells. Both 3,6′-DP and 1,6′-DP increased cell survival and reduced neuroinflammatory-related cytokines in these in vitro cultures after challenge with α-synuclein and LPS, respectively. We therefore conclude that 3,6′-DP and 1,6′-DP may have translational potential for stroke therapy. 

Our current studies follow up on former ones indicating that 3,6′-DP possesses a favorable CNS MPO (multiparameter optimization) score of 5.5 that is predictive for an agent possessing desirable drug-like properties for neurological action [27]. It appears to readily enter the brain (brain/plasma concentration ratio 0.8) [28] and has demonstrated efficacy in rats subjected to traumatic brain injury [28,46]. In these TBI studies, 3,6′-DP significantly mitigated microgliosis and astrogliosis in the surrounding brain, and reduced neuronal cell loss, injury contusion volume and behavioral impairments at both 24 h and 7 days post-injury, proving more effective than equimolar Pomalidomide [28,46]. 

The non-cereblon-mediated actions of IMiDs to lower TNF-α are considered mediated via translational regulation at the level of the 3′-untranslated region of TNF-α mRNA, which results in accelerated mRNA degradation and lower TNF-α protein generation [49]. This has been found across human monocytes challenged with LPS [50], as well as human tuberculosis patients’ alveolar macrophages stimulated by LPS [51] and human alveolar macrophages challenged with LPS from patients with interstitial lung disease [52]. This, likewise, seems to be pertinent to brain resident microglial cells following an inflammatory challenge (α-synuclein in the current study). Such actions appear to block NF-κb transcription factors [53] and lower the generation of other pro-inflammatory cytokines to thereby blunt the inflammatory response [53]. 

It is widely considered that TNF-α plays both a critical and initial role in the pathophysiology of stroke and provides both neurotoxic and neuroprotective effects [54,55]. A rise in TNF-α is one of the earliest events to occur as part of an inflammatory response to ischemic brain damage, peaking between 12–24 h and contributing to triggering the cascade of other inflammatory components in blood, brain and CSF in stroke animal models [54]. Elevated TNF-α has similarly been reported in stroke patients as early as 6–12 h post-symptom onset [56], rising between 24 and 48 h and gradually declining from 72–144 h [57]. Several studies have reported that higher post-stroke TNF-α levels are associated with poorer long-term outcomes or larger infarct size [43]. If true, strategies focused on TNF-α that dampen the inflammatory response may prove beneficial. TNF-α antibodies and TNF-α binding proteins have demonstrated efficacy in preclinical stroke models, including MCAo [55,58]. However, such protein strategies have limited brain access, although BBB leakage can occur following ischemic stroke. Nevertheless, small molecular weight drugs that readily enter the brain and can lower systemic and central TNF-α levels may better translate into humans. Mouse TNF-α knockout studies demonstrate that a complete lack of TNF-α results in a larger infarct volume and greater behavioral deficits [54], indicating that some level of TNF-α is essential and that time-dependently blunting rather than removing the inflammatory event is an approach to follow.

The question arises as to the relevance of plasma cytokine levels measured here to stroke development. Stroke has been traditionally defined as an acute neurological disorder [59]. However, over the last decade, the chronic “neurodegenerative-like” pathological manifestation of stroke has been recognized. Chronic neuroinflammation closely accompanies the disease progression [60]. Moreover, while stroke largely manifests brain pathology with altered neurological function, the key role of the periphery in the disease evolution, in particular, immune and inflammatory responses from the spleen and gut, have been recently documented [61,62]. The splenic immune response exacerbates stroke symptoms in both animals and patients [63]. Similarly, upregulation of inflammation-associated microbiomes in the gut approximates worsening of stroke progression [61,64]. Together, these findings suggest a major involvement of the systemic immune system in stroke.

The key role of the peripheral immune system in the pathological symptoms of stroke supports the notion that sequestration of deleterious brain immune responses may benefit from regulation of the systemic immune signaling pathways representing novel therapeutic targets for stroke [65,66]. Recent studies show that targeting regenerative medicine-based therapies to peripheral organs, such as the spleen and the gut, reduces behavioral and histological deficits associated with experimental stroke [67,68]. Accumulating evidence also indicates that MHC molecules may attenuate stroke-induced neuroinflammation by tapering harmful immune and inflammatory alterations in the periphery [67,68]. Thus, recognizing that stroke entails both central and peripheral alterations in the immune response warrants greater understanding of systemic immune signaling pathways to fully explore the pathology and treatment of stroke [7]. In this context, potentially less toxic thalidomide/pomalidomide analogs with the ability to lower TNF-α and dampen the immune response both systemically and within the central nervous system may hold potential as a treatment strategy for stroke. In this light, 3,6′- and 1,6′-DP appear to effectively fulfill this role, meet initial FDA guidelines for consideration of translation and, hence, warrant further development. Having demonstrated a clear initial signal of efficacy in male rats with MCAo-induced cerebral ischemia, future studies should incorporate females and evaluate longer durations and broader measures of response. This could include markers of cerebral edema as well as classical potential adverse actions associated with the wide and successful prior use of the IMiD therapeutic class in the cancer field. 

## 5. Conclusions

The novel pomalidomide analogs 3,6′-DP and 1,6′-DP effectively mitigate inflammation across cellular models of α-synuclein and LPS challenge as well as male rat models of LPS and MCAo challenge. Like pomalidomide, both agents reduce ischemic stroke infarct volume and associated behavioral impairment. However, although, similar to pomalidomide, 3,6′-DP potently binds to cereblon, and it importantly does not appear to induce the ubiquitination of key downstream neo-substrates (Ikaros, Aiolos and SALL4) associated with the anticancer/teratogenicity of the IMiD drug class [16,17,27]. 3,6′-DP and 1,6′-DP lack activity in essential gatekeeper genotoxicity and hERG assays critical to human translation, and, in the light of their clear efficacy signal in the current study, they thus warrant further evaluation and development in preclinical animal models as candidate drugs to mitigate inflammation associated with excessive pro-inflammatory cytokine generation. In the light of the current data and recent favorable results of 3,6′-DP activity in a preclinical model of Alzheimer’s disease [69], future studies are focused towards evaluating the acute and chronic tolerability/toxicity of 3,6′- and 1,6′-DP in both male and female large and small animal species as a segue to potential human studies. 

## 6. Patents

3,6′-DP and 1,6′-DP are protected under US Patent 8,927,725 and associated patents.

## Figures and Tables

**Figure 1 pharmaceutics-14-00950-f001:**
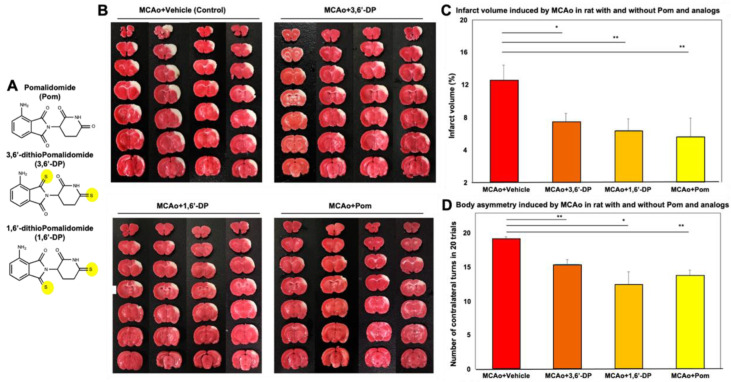
Infarct volume and body asymmetry quantification in the rat MCAo model. (**A**) Chemical structures of pomalidomide, 3,6′-DP and 1,6′-DP. (**B**) Representative coronal brain sections, stained with 2% 2,3,5-triphenyltetrazolium chloride (TTC) solution 24 h post-MCAo/reperfusion. Animals were divided across vehicle and three treatment groups: MCAo + vehicle, MCAo + 3,6′-DP, MCAo + 1,6′-DP and MCAo + POM (pomalidomide). TTC is a colorless water-soluble dye that is reduced by the mitochondrial enzyme succinate dehydrogenase of living cells into a water-insoluble, light-sensitive compound (formazan) that turns healthy/normal tissue a deep red color. In contrast, damaged/dead tissue remains white, showing the absence of living cells and thereby indicating the infarct region [44]. (**C**) Quantitative infarct volumes of groups treated with/without drug. (**D**) Body asymmetry was evaluated by “elevated body swing test” [33,34] to quantify contralateral turns across groups (non-MCAo rodents: 10 turns/20 trials; making equal numbers of contra- and ipsilateral turns). In contrast, a rodent with a significant one-sided lesion in a motor-related area may score 20 contralateral turns/20 trials). Data are mean + standard error of the mean (SEM, *n* = 4/group), Dunnett’s *t*-test * *p* < 0.05, ** *p* < 0.01 versus MCAo + vehicle group. Image analysis software (Image J, NIH).

**Figure 2 pharmaceutics-14-00950-f002:**
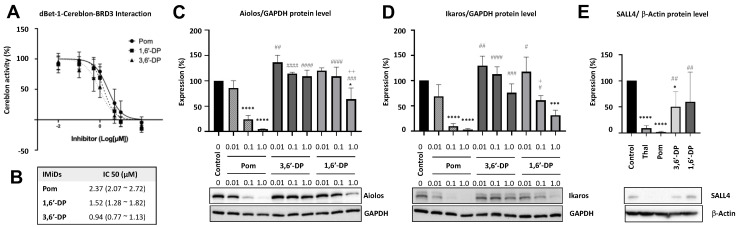
3,6′-DP, 1,6′-DP and pomalidomide potently bind cereblon, but 3,6′-DP does not lower downstream neo-substrates, epitomized by Ikaros, Aiolos and SALL4. Binding (**A**) of pomalidomide (POM), 3,6′-DP and 1,6′-DP to cereblon provided IC_50_ values (**B**) of 2.37, 0.94 and 1.52 μM, respectively. Degradation of downstream neo-substrates Aiolos and Ikaros were evaluated in human MM.1S cells. (**C**) Amounts of 0.1 and 1.0 μM POM dramatically lowered Aiolos and Ikaros protein levels in human MM1S cells (normalized to GADPH), these same concentrations of 1,6′-DP less substantially lowered Aiolos (**C**) and Ikaros (**D**). Notably, 3,6′-DP had no significant action on protein degradation of Aiolos and Ikaros. (**E**) SALL4 protein levels were quantified in H9 cells challenged with thalidomide, pomalidomide, 1,6′- and 3,6′-DP (3 μM). All, with the exception of 3,6′-DP, significantly lowered SALL4 (normalized to β-actin). * *p* < 0.05, *** *p* < 0.001, **** *p* < 0.0001: treatments vs. control (i.e., DMSO/vehicle). # *p* < 0.05, ## *p* < 0.01, ### *p* < 0.001, #### *p* < 0.0001: treatments vs. POM of same concentration. + *p* < 0.05, ++ *p* < 0.01: 1,6′-DP vs. 3,6′-DP at same concentration (values: mean ± SEM, *n* = 4/group).

**Figure 3 pharmaceutics-14-00950-f003:**
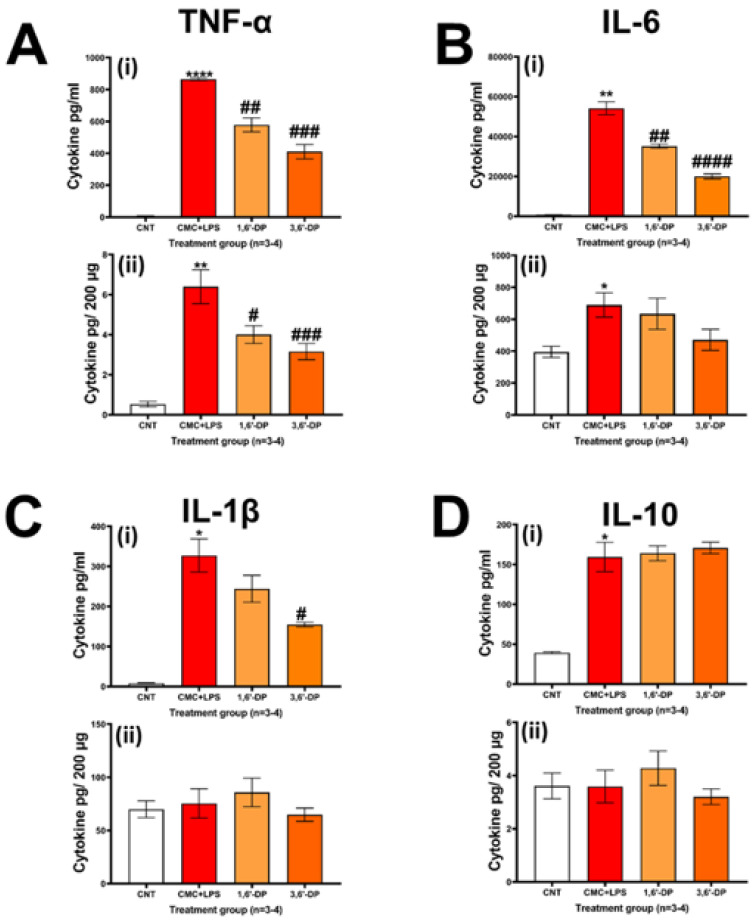
Systemic 3,6′-DP and 1,6′-DP mitigate plasma and brain inflammation markers in the LPS rodent inflammation model. Animals were pre-treated with vehicle or drugs 1 h prior to LPS (1 mg/kg, i.p.) or LPS vehicle (saline). Animals were euthanized 4 h later, and plasma and brain (hippocampus) collected. This brain area was selected as it possesses a high baseline metabolic activity, and thus the hippocampal CA1 region is extremely susceptible to reduced oxygen and glucose and neuroinflammation [47]. (**A**(**i**))**:** LPS induced a marked elevation in plasma TNF-α protein levels, and both drugs significantly attenuate this LPS effect. (**B**(**i**))**:** Similarly, LPS elevated levels of IL-6 protein, and both drugs significantly attenuated this LPS action. (**C**(**i**))**:** LPS induced a significant increase in plasma IL-1β protein, which was significantly attenuated by 3,6′-DP. (**D**(**i**))**:** LPS induced an elevation in IL-10 protein levels, neither drug had any effect on plasma IL-10 levels. Hippocampal levels of LPS-induced TNF-α (**A**(**ii**)) and IL-6 (**B**(**ii**)) were significantly elevated vs. control (CNT: vehicle + saline). Both drugs significantly reduced brain TNF-α levels, and a trend for reduction in IL-6 did not attain statistical significance. Actions of LPS administration on tissue levels of IL-1β and IL-10 were not significantly different from control levels (**C**(**ii**)**,D**(**ii**)). * *p* < 0.05, ** *p* < 0.01, **** *p* < 0.0001: control vs. LPS alone (red bars, unpaired Student’s *t*-test). # *p* < 0.05, ## *p* < 0.01, ### *p* < 0.001, #### *p* < 0.0001: drugs + LPS vs. LPS alone (one-way ANOVA and Dunnett’s multiple comparison test). Data are mean ± SEM, *n* = 3–4/group.

**Figure 4 pharmaceutics-14-00950-f004:**
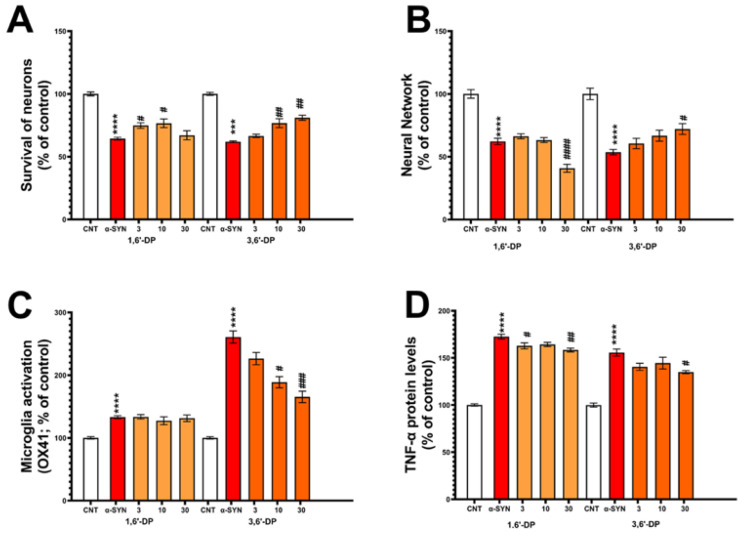
3,6′-DP and 1,6′-DP attenuate losses of dopaminergic cells, neurite networks and reduce levels of activated microglia and TNF-α protein induced by α-synuclein (α-SYN) challenge. Primary co-cultures of dopaminergic cells were preincubated with drugs (3–30 µM, 1 h) before a 72 h α-SYN (250 nM). α-SYN-challenged cells in presence of vehicle (red bar: DMSO) displayed significant reductions in (**A**) survival of neurons and (**B**) neurite networks, significant elevations in levels of (**C**) activated microglial cells and (**D**) TNF-α protein. 3,6′-DP and 1,6′-DP dose-dependently mitigated the α-SYN actions across all measures. *** *p* < 0.001, **** *p* < 0.0001: control vs. α-SYN alone (red bars, unpaired Student’s *t*-test). # *p* < 0.05, ## *p* < 0.01, ### *p* < 0.001, #### *p* < 0.0001: compounds vs. α-SYN alone (one-way ANOVA and Dunnett’s multiple comparison test). Data are mean ± SEM, *n* = 4–6/group.

**Figure 5 pharmaceutics-14-00950-f005:**
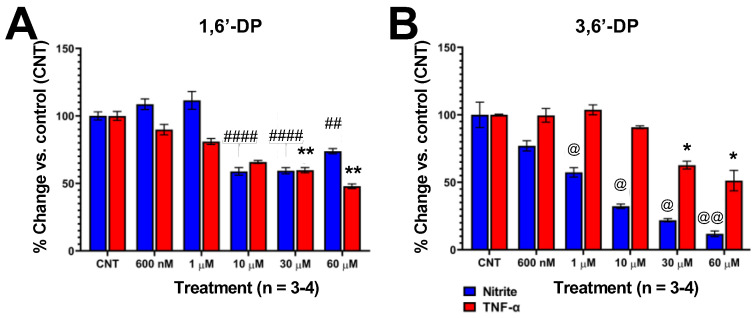
3,6′-DP and 1,6′-DP attenuate LPS-induced elevations in NO^2−^ and TNF-α protein levels in RAW 264.7 cells. Effects of (**A**) 1,6′-DP and (**B**) 3,6′-DP (600 nM–60 µM) were assessed on RAW 264.7 cell viability (not shown), NO^2−^ and TNF-α protein levels in presence of LPS (60 ng/mL). 3,6′-DP concentration-dependently (1–60 µM) reduced levels of NO^2−^ and TNF-α (30–60 µM). 1,6′-DP significantly lowered NO^2−^ at 10 µM with no further reductions at 30–60 µM. 1,6′-DP (30–60 µM) induced significant reductions in TNF-α. ## *p* < 0.01, #### *p* < 0.0001: compounds + LPS vs. LPS alone (CNT) (one-way ANOVA and Dunnett’s multiple comparison test). * *p* < 0.05, ** *p* < 0.01: compounds + LPS vs. LPS alone (CNT) (Kruskal–Wallis test followed by a Dunn’s multiple comparison test). @ *p* < 0.05, @@ *p* < 0.01: compounds + LPS vs. LPS alone (CNT) (Brown–Forsythe and Welch’s ANOVA test + Dunnett’s T3 multiple comparison test). Data are mean ± SEM, *n* = 3–4/group.

**Table 1 pharmaceutics-14-00950-t001:** Pomalidomide analogs reduce cytokine-related inflammation after MCAo injury.

N = 4	Sham Control	MCAo + Vehicle	MCAo + 3,6′-DP	MCAo + 1,6′-DP	MCAo + POM
IL-1β (pg/)mL	6.58 ± 2.08	10.20 ± 0.76	4.06 ± 0.95 **	2.93 ± 0.21 **	4.76 ± 1.01 *
TNF-α (pg/)mL	4.08 ± 0.88 *	7.62 ± 1.04	3.39 ± 0.15 **	<2.22	2.59 ± 0.14 **
IL-10 (pg/)mL	14.28 ± 3.16	12.51 ± 0.76	26.49 ± 0.86 **	11.59 ± 3.78	16.62 ± 3.21

Vehicle or pomalidomide (POM 20 mg/kg, i.p.), 3,6′-DP and 1,6′-DP (21.25 mg/kg, i.p.) were administered to animals challenged with MCAo 60 min after occlusion and reperfusion. Plasma samples were obtained 24 h later, and IL-1β, IL-10 and TNF-α levels were quantified (Milliplex platform). All outcomes were compared to MCAo + vehicle group (Dunnett’s *t*-test: * *p* < 0.05, ** *p* < 0.01 versus MCAo + vehicle group). TNF-α in the MCAo + 1,6′-DP group fell below the assay minimum detection level (2.22 pg/mL). The standard ranges for the assays were IL-1β: 2.4–10,000 pg/mL TNF-α: 2.4–10,000 pg/mL and IL-10: 7.3–30,000 pg/mL. Values are mean + SEM (*n* = 4/group); vehicle: DMSO alone.

## Data Availability

All data is available on request from the corresponding authors.

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
