# Peer review of "3,6′- and 1,6′-Dithiopomalidomide Mitigate Ischemic Stroke in Rats and Blunt Inflammation"

_pharmaceutics, 2022, doi:10.3390/pharmaceutics14050950_

Round 1
Reviewer 1 Report
The manuscript by Tsai et al. investigates the effects of pomalidomide derivatives on neuroinflammation in several in vivo and in vitro models. Overall, the data convincingly show that these drugs are able to effect the production of inflammatory cytokines, but there are several issues with the submitted manuscript.
In Table 1, it says a t-test was performed in the legend but ANOVA was used in the Methods section. Why was a t-test run for Table 1 when there are 5 treatment groups?
Why were brain cytokine levels measured in the LPS model and not the MCAO model? It seems like they would be equally important in both models. Why was only the hippocampus measured for “brain” samples? Some more justification as to why there is no measurement of brain cytokine levels in the MCAO model and why only the hippocampus was measured in the LPS model would be helpful.
Graphs in figure 3 and 4 are too small and I am unable to evaluate them. Please make these figures larger or the text labels easier to read.
How do the authors know they were measuring brain cytokine levels instead of plasma levels from the blood vessels in the brain? What was done to distinguish the measurements inside the brain versus inside the cerebral vasculature?
Author Response
Reviewer 1
The manuscript by Tsai et al. investigates the effects of pomalidomide derivatives on neuroinflammation in several in vivo and in vitro models. Overall, the data convincingly show that these drugs are able to effect the production of inflammatory cytokines, but there are several issues with the submitted manuscript.
- In Table 1, it says a t-test was performed in the legend but ANOVA was used in the Methods section. Why was a t-test run for Table 1 when there are 5 treatment groups?
Response: In Table 1 the abilities of Pomalidomide and novel analogs 3,6’-DP and 1,6’-DP were evaluated to reduce MCAo-induced cytokine-related inflammation. As noted by Reviewer 1, there are 5 groups: (i) a Sham Control group (administered vehicle and without MCAo injury, (ii) a MCAo injury + vehicle group, (iii) a MCAo injury + 3,6’-DP group, (iv) a MCAo injury + 1,6’-DP group and (v) a MCAo injury + Pom group. In our statistical analysis, a Dunnet’s t-test is used to compare groups (i), (iii), (iv) and (v) vs. the (ii) MCAo injury + vehicle group.
The Dunnett's t test specifically allows comparison of ‘one’ group (i.e., a single group – specifically group (ii) in our Table 1) with the other groups; thereby, addressing a special case among multiple comparison problems.
More generally, when an ANOVA test shows significant findings, it doesn’t report which pairs of means are different. A Dunnett’s t-test can be used after the ANOVA has been run to identify the pairs with significant differences from a chosen selected group (i.e., group (ii) in Table 1).
We believe that the selected statistical evaluations in our manuscript are the correct one, and the explanation above provides insight into why the Dunnett’s test was appropriately chosen.
- Why were brain cytokine levels measured in the LPS model and not the MCAO model? It seems like they would be equally important in both models. Why was only the hippocampus measured for “brain” samples? Some more justification as to why there is no measurement of brain cytokine levels in the MCAO model and why only the hippocampus was measured in the LPS model would be helpful.
Response: Both routinely and classically, the major measure in brain ischemic stroke studies is the quantification of ‘infarct volume’ together with a behavioral measure (in our study: body asymmetry quantification). The most used technique to quantify infarct volume is from coronal brain sections, stained with 2% 2,3,5-Triphenyl tetrazolium chloride (TTC) solution 24 hr post-MCAo/reperfusion, and is the technique we used in our rat MCAo study – and is shown in Figure 1.
TTC is a colorless water-soluble dye that is reduced by the mitochondrial enzyme succinate dehydrogenase in living cells into a water-insoluble, light sensitive compound (formazan) that turns healthy/normal tissue a deep red color. In contrast, damaged/dead tissue remains white showing the absence of living cells, and thereby indicating the infarct region (Bederson, J. B., et al. Stroke 17, 1304–08, 1986).
Quantification of proinflammatory cytokines in brain by ELISA (Mesoscale Discovery) is required to be undertaken on brain samples immediately dissected and frozen to -80 C. This procedure cannot be undertaken on brain tissue prepared and stained with TTC. As a consequence, proinflammatory cytokines in brain were measured from immediately frozen brain samples deriving from a classical model of neuroinflammation (the LPS model).
The brain area evaluated was the hippocampus – as an area with a high baseline metabolic activity, the hippocampal CA1 region is extremely susceptible to reduced oxygen and glucose and neuroinflammation (Marcoux FW, et al. Stroke. 13:339–46, 1982) – and this is the reason that we selected the hippocampus as the brain region to evaluate.
- Graphs in Figures 3 and 4 are too small and I am unable to evaluate them. Please make these figures larger or the text labels easier to read.
Response: We agree and have increased the size of these Figures in our newly modified manuscript, in line with the Reviewer’s suggestion
- How do the authors know they were measuring brain cytokine levels instead of plasma levels from the blood vessels in the brain? What was done to distinguish the measurements inside the brain versus inside the cerebral vasculature?
Response: We have previously cleared blood volume by perfusion of animals with cold isotonic saline solution, and found similar brain levels of cytokines.
Reviewer 2 Report
Respected Sir
Many thanks for your efforts done during this valuable work.
Author Response
Reviewer 2
- Many thanks for your efforts done during this valuable work.
Response: the authors thank Reviewer 2 for the time taken to kindly peer-review our manuscript
We have used 'spell check' to identify some typos and misspellings in our original manuscript and have corrected these in our revised version. Additionally, detailed Methods are provided in our Supplemental File.
Reviewer 3 Report
The authors should explain why only male rats were used.
The authors should explain why different strains of rats were used
(2.2.1. MCAo procedure: Transient MCAo was performed in male Sprague Dawley rats...
2.4.1. Lipopolysaccharide (LPS) inflammatory challenge: Male Fischer rats...)
References should be provided for statements in lines 255-262: "Ischemic stroke triggers a vigorous inflammatory response initiated within minutes (Ref?) and persisting for days.
Minutes (Ref?) to hours after stroke, the acute phase of stroke leads to production of pro-inflammatory cytokines, particularly IL-1 and TNF-α, which propagate the neuroinflammatory response through activation of danger signals."; 431-432: "An important concomitant effect of stroke in both humans and preclinical models is neuroinflammation, especially in the penumbra (Ref?) region." and 490: "Stroke has been traditionally defined as an acute neurological disorder."
Author Response
Reviewer 3
- The authors should explain why only male rats were used.
Response: Gender differences are known in the response of rodents to MCAo, as well as to other brain insults (Baskerville TA, et al., J Cereb Blood Flow Metab. 36: 381-6, 2016; and El-Hakim Y, et al., Biol Sex Differ. 12: 14, 2021). It is well established that stroke incidence in young- to middle-aged adults is lower in women than men. This trend reverses later in life, particularly when women enter their postmenopausal stage (Kim T, et al., Neurochem Int 127:22–8, 2019). Likewise, young female rodents appear to be more resistant to ischemic brain injury than age-matched males (Alkayed NJ, et al. Stroke 29:159–66, 1998. discussion 166; Vannucci SH et al., J Cereb Blood Flow Metab 21:52–60, 2001). Studies in ovariectomised rodents have demonstrated that estrogens provide significant neuroprotection to counter ischemic stroke damage in female animals (Alkayed et al., 1998 ibid; Vannucci et al., 2001 ibid; Suzuki S, et al., J Comp Neurol; 500:1064–75, 2007). In contrast, estrogen is reported to enlarge infarct volume in aged female rodents . In summary, these studies suggest that estrogen protects the younger brain but exacerbates the damage in the ageing brain, after ischaemic stroke (Gordon KB, et al., Brain Res 1036:155–62, 2005; Carswell HV, et al., J Cereb Blood Flow Metab 24: 298–304, 2004). To avoid the confounds associated with estrogen generation in young female rats, ovariectomizing animals or aging them to a postmenopausal state, we performed and report our initial studies in young adult male rats (we have described this in lines 148-159 in our revised manuscript). Furthermore, we have modified the Discussion of our manuscript to reflect this and indicate that in the light of positive actions in males, future studies in females are now warranted (lines 542-544, and 562).
The authors should explain why different strains of rats were used.
Response: we selected the strains of rats commonly used in the literature in the models studied here.
References should be provided for statements in lines 255-262: "Ischemic stroke triggers a vigorous inflammatory response initiated within minutes (Ref?) and persisting for days.
Minutes (Ref?) to hours after stroke, the acute phase of stroke leads to production of pro-inflammatory cytokines, particularly IL-1b and TNF-α, which propagate the neuroinflammatory response through activation of danger signals.";
431-432: "An important concomitant effect of stroke in both humans and preclinical models is neuroinflammation, especially in the penumbra (Ref?) region." and 490: "Stroke has been traditionally defined as an acute neurological disorder."
Response: We agree with Reviewer 3, and have added appropriate references with numbers cited below and added to the manuscript:
“Ischemic stroke triggers a vigorous inflammatory response initiated within minutes and persisting for days [45,56]. After ischemic stroke a secondary immune response occurs that include glial activation and release of cytokines and chemokines. To study the effects of Pomalidomide and analogs in ischemic stroke-induced inflammation, we investigated cytokine levels after post-treatment with Pomalidomide and analogs in MCAo.
Minutes to hours after stroke, the acute phase of stroke leads to production of pro-inflammatory cytokines, particularly IL-1b and TNF-α, which propagate the neuroinflammatory response through activation of danger signals [7,45].”
“An important concomitant effect of stroke in both humans and preclinical models is neuroinflammation, especially in the penumbra region [3-8,10-12]. The clinically approved IMiDs, thalidomide and pomalidomide, exhibit potent anti-inflammatory actions [27].”
"Stroke has been traditionally defined as an acute neurological disorder [59]."
Finally, we have added approx. 13 additional References to our revised manuscript to ensure that it is fully and appropriately referenced.
Reviewer 4 Report
In this work, the anti-inflammatory effect of two new drugs pomalidomide derivatives, 3,6'-dithiopomalidomide (3,6'-DP) and 1,6'-dithiopomalidomide (1,6'-DP) was evaluated in comparison with pomalidomide. It has been shown that the drugs have an anti-inflammatory effect mainly due to a decrease in the level of TNF-α and IL1-β and do not have a teratogenic effect.
Overall, the core conclusions are supported by the evidence, and the data presented represent an advance in the current literature. But I have some comments: 1. The authors claim that the drugs are able to reduce infarct volume and functional deficit (Figure 1.). However, it is known that in the MCAo model, the volume of the infarct varies greatly, as well as the functional parameters, so the use of only 4 animals per group is clearly not enough. In fact, only these two tests indicate the neuroprotective effect of drugs. 2. The "body asymmetry test" is done 24 hours after the MCAo. It should be shown if the single use of this test in the acute period after stroke is sufficient to assess the neuroprotective effect of anti-inflammatory drugs. Possibly according to other studies. 3. It would be good to increase the number of tests indicating the neuroprotective effect of the studied drugs. 4. One of the main known limitations of pomalidomide analogs is the increased incidence of thromboembolic complications in patients with multiple myeloma, and the idea of recommending pomalidomide-based drugs for the treatment of ischemic stroke seems doubtful. In the discussion, it is necessary to discuss the effect of the studied drugs on the formation of blood clots and recurrent stroke. 5. Also in the discussion I would like to see information on whether drugs based on pomalidomide, which have anti-inflammatory effects, can reduce cerebral edema after stroke. But in general, the comments made do not reduce the value and relevance of the work.Author Response
Reviewer 4
In this work, the anti-inflammatory effect of two new drugs pomalidomide derivatives, 3,6'-dithiopomalidomide (3,6'-DP) and 1,6'-dithiopomalidomide (1,6'-DP) was evaluated in comparison with pomalidomide. It has been shown that the drugs have an anti-inflammatory effect mainly due to a decrease in the level of TNF-α and IL1-β and do not have a teratogenic effect.
Overall, the core conclusions are supported by the evidence, and the data presented represent an advance in the current literature. But I have some comments:
- The authors claim that the drugs are able to reduce infarct volume and functional deficit (Figure 1.). However, it is known that in the MCAo model, the volume of the infarct varies greatly, as well as the functional parameters, so the use of only 4 animals per group is clearly not enough. In fact, only these two tests indicate the neuroprotective effect of drugs.
Response: Whereas we understand the Reviewer’s perspective, we respectfully disagree with the Reviewer. Figure 1 demonstrates a “statistically significant” reduction in infarct volume (at the p<0.05 and p<0.01 level) of the treatments vs. the MCAo vehicle group with the number of animals utilized (n=4/group) and, likewise, there is a statistically significant decline in the postural asymmetry measure associated with this reduction. This animal number was selected based on prior studies (see Methods) – and since a statistically significant drug-induced mitigation was achieved, the use of additional animals is not scientifically warranted. Additionally, our research institutions strongly adhere to the “Three Rs principles for humane animal research” (Hubrecht RC, Carter E. The 3Rs and Humane Experimental Technique: Implementing Change. Animals (Basel). 2019; 9(10):754): Replacement, Reduction and Refinement (Russell WMS, Burch RL. The Principles of Humane Experimental Technique. London, UK: Universities Federation for Animal Welfare; 1959). In relation to ‘Reduction’ - referring to any strategy that will result in fewer animals being used to obtain sufficient data to answer the research question, or in maximizing the information obtained per animal and thus potentially limiting or avoiding the subsequent use of additional animals, without compromising animal welfare.
In synopsis, our institutions would consider it ‘unethical’ for my colleagues and I to undertake studies on additional animals, as we already have achieved statistical significance with the animals presently dedicated to the study. In this light, it is very unlikely that our institutions would approve our potential request to undertake studies to increase the current animal numbers in our animal groups relating to Fig. 1.
- The "body asymmetry test" is done 24 hours after the MCAo. It should be shown if the single use of this test in the acute period after stroke is sufficient to assess the neuroprotective effect of anti-inflammatory drugs. Possibly according to other studies.
Response: These studies – demonstrating a close relation between body asymmetry and infarct volume have been undertaken by Borlongan – who originally developed the elevated body swing test (EBST) that is now widely used as a quantitative measure of asymmetry (see: Borlongan C.V.; Sanberg P.R. J Neurosci. 1995, 15, 5372-5378; Borlongan C.V.; Hida H.; Nishino H. Neuroreport. 1998, 9, 3615-3621).
One of the most significant symptoms of stroke is sensorimotor asymmetry, which can be assessed by several behavioral tests, - particularly by EBST. Borlongan et al., have previously reported a highly significant correlation between volume of infarction induced by proximal MCAo and body asymmetry measured by EBST (Borlongan, Hida et al., 1998). A wealth of studies by Borlongan and colleagues have evaluated drug classes to mitigate this (including agents to impact neuroinflammation). The value of this useful tool has been confirmed in Shen and Wang (J Neurosci Methods. 2010 Feb 15; 186(2): 150–154.) and in numerous other publications. – We have modified our revised manuscript to state the above (lines 455-461).
- It would be good to increase the number of tests indicating the neuroprotective effect of the studied drugs.
Response: We agree with the Reviewer, and consider that such studies would be valuable in a separate followup study – focused on the agent 3,6’-dithioPomalidomide (as it lacks downstream actions associated with cereblon binding – and may hence show greater tolerability in relation to Pomalidomide). Having demonstrated initial activity in the present paper, in keeping with the “Three Rs” – we would seek to undertake a dose-response study (evaluating half-log reductions in dose) over 24 hr and a greater duration (7 and 30 days) with a broader number of measures (behavioral, biochemical and measures of edema).
We have modified our Discussion to state the above (lines 543-5460
- One of the main known limitations of pomalidomide analogs is the increased incidence of thromboembolic complications in patients with multiple myeloma, and the idea of recommending pomalidomide-based drugs for the treatment of ischemic stroke seems doubtful. In the discussion, it is necessary to discuss the effect of the studied drugs on the formation of blood clots and recurrent stroke.
Response: We agree with the Reviewer that Pomalidomide use is associated with an increased risk of thromboembolic complications when used in the treatment of multiple myeloma (particularly for over a month duration).
The risk of thromboembolism is approx. 7–10 times higher in patients with cancer as compared with the normal population. The risk is considerably higher still (28 times) in patients with hematologic cancer and multiple myeloma [Blom et al. 2005]. The pathophysiology of thrombosis in cancer is a complex process – and does not fit well in a study in relation to ischemic stroke and IMiDs. The interaction between malignant cells and monocyte/macrophage cells stimulates the release of TNF, IL-1 and IL-6, all of which can induce endothelial damage. Interactions between tumor cells and macrophages also activates platelets, factor XII and X. Cysteine protease and tissue factor are highly expressed in cancer cells, have procoagulant activity and can directly activate factor X and VII [Prandoni et al. 2005]. There appears to be a dramatic increase in the risk of developing thromboembolic events during multiple myeloma treatment with thalidomide–anthracycline combination regimens, [Barlogie et al. 2006].
How the above risks of thromboembolism would relate to an individual without multiple myeloma and/or hematological cancer that may have a cerebral ischemic event and be treated with a single dose or short duration dose of a IMiD (in the absence of combination with an anticancer anthracycline) is unknown – and, my colleagues and I believe that the introduction of this topic in the Discussion of our manuscript would be confusing to most readers (certainly, should our strategy of potential IMiD use in cerebral ischemia ultimately move to a clinical stage, the FDA and alike regulatory authorities would evaluate potential thromboembolism potential – as this is known in the use of this drug class in multiple myeloma).
We have modified our revised manuscript to clearly state the importance of evaluating adverse actions of the IMiD therapeutic class in relation to what is known from its wide and successful application in the field of cancer treatment (line 546)
- Also in the discussion I would like to see information on whether drugs based on pomalidomide, which have anti-inflammatory effects, can reduce cerebral edema after stroke. But in general, the comments made do not reduce the value and relevance of the work.
Response: We agree with the Reviewer that measures of cerebral edema would be most useful to evaluate in a follow up study – directed at 3,6’-dithioPomalidomide (as discussed in response to Question 3, above). We have added a line in this regard in the Discussion as a future direction (line 545).
In addition to the above – we have again ‘spell checked’ our manuscript and made minor corrections throughout. Detailed additional Materials and Methods are available in the Supplemental Information file.